# Sparse synaptic connectivity is required for decorrelation and pattern separation in feedforward networks

N. Alex Cayco-Gajic[1], Claudia Clopath[2] & R. Angus Silver [1]

Pattern separation is a fundamental function of the brain. The divergent feedforward networks thought to underlie this computation are widespread, yet exhibit remarkably similar sparse synaptic connectivity. Marr-Albus theory postulates that such networks separate overlapping activity patterns by mapping them onto larger numbers of sparsely active neurons. But spatial correlations in synaptic input and those introduced by network connectivity are likely to compromise performance. To investigate the structural and functional determinants of pattern separation we built models of the cerebellar input layer with spatially correlated input patterns, and systematically varied their synaptic connectivity. Performance was quantified by the learning speed of a classifier trained on either the input or output patterns. Our results show that sparse synaptic connectivity is essential for separating spatially correlated input patterns over a wide range of network activity, and that expansion and correlations, rather than sparse activity, are the major determinants of pattern separation.

[1] Department of Neuroscience, Physiology and Pharmacology, University College London, London WC1E 6BT, UK. [2] Bioengineering Department, Imperial College London, London SW7 2AZ, UK. Correspondence and requests for materials should be addressed to R.A.S. (email: a.silver@ucl.ac.uk)

The ability to distinguish similar, yet distinct patterns of sensory input is a core feature of the nervous system. Pattern separation underlies such everyday activity as recognizing faces and distinguishing odors. Early theoretical work by Marr and Albus[1, 2] showed that divergent excitatory feedforward networks can separate patterns of neuronal activity by projecting them onto a larger population (called 'expansion recoding') and reducing the fraction of neurons active, forming a 'sparse' population code in which the overlap between distinct neuronal firing patterns is reduced[3–7]. Divergent feedforward networks, thought to be involved in pattern separation, are widespread in the nervous system of both vertebrates and invertebrates, including the olfactory bulb[8, 9], mushroom body[10, 11], dorsal cochlear nucleus[12] and hippocampus[13, 14]. But perhaps the most well studied example is the input layer of the cerebellar cortex, which combines many different types of sensory modalities and motor command signals[15]. The input layer of the cerebellar cortex has an evolutionarily conserved network structure, in which granule cells receive 2–7 synaptic inputs, with the claw-like ending of each dendrite innervating a different

mossy fibre[15]. Interestingly, other divergent feedforward networks also have relatively few synapses: granule cells in the dorsal cochlear nucleus have 2–3 dendrites[16] while Kenyon cells in the fly olfactory system have around 7 synaptic inputs[17]. This raises the question of why the synaptic connectivity of these networks is so similar. Recent studies have provided a potential solution, showing that having few synaptic inputs per granule cell provides an optimal solution to a trade-off between information transmission and sparsening population activity[18], and optimizes associative learning in feedforward networks with sparse coding levels[19]. However, several key questions remain regarding how the structure of feedforward networks supports pattern separation.

Marr-Albus theory posits that sparse coding and expansion recoding together reduce pattern overlap[1, 2, 7], while more recent work highlights the importance of input decorrelation[8, 10, 11, 20–24]. However, it is not known how much each factor separately contributes to pattern separation and learning, or how they depend on network structure. In addition, theoretical studies have generally focused on idealized,

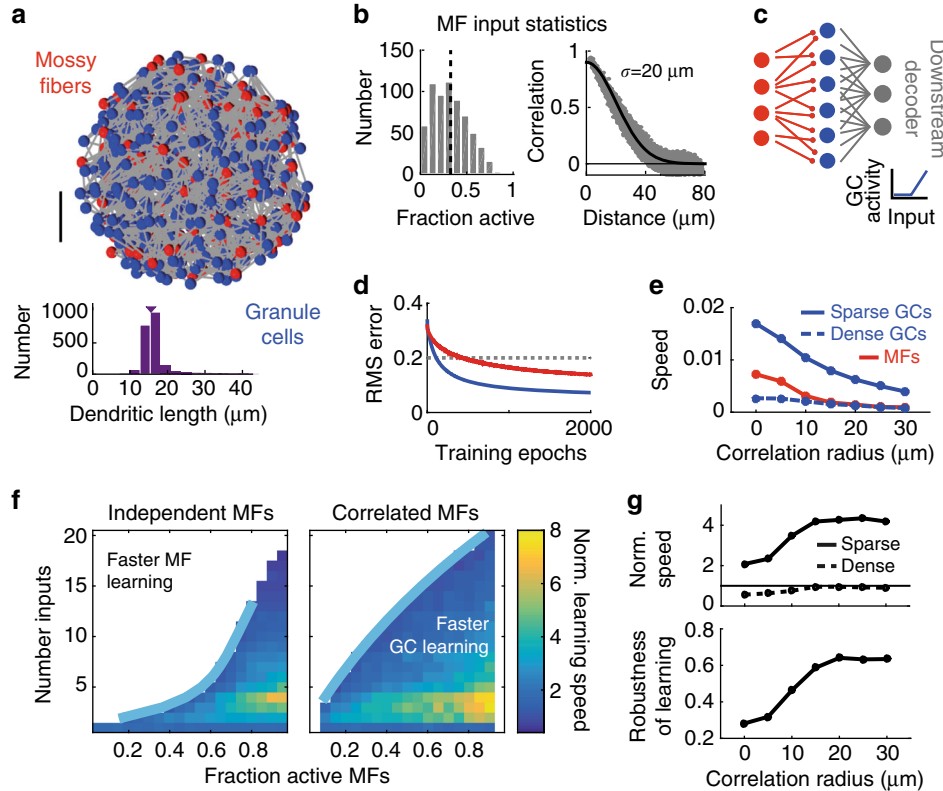

**Fig. 1** A simple feedforward model of the cerebellar input layer with sparse, but not dense, synaptic connectivity speeds learning. **a** Top: Anatomically constrained 3D model of cerebellar input layer. Positions of Granule Cells (GCs, blue) and Mossy Fibers (MFs, red) within an 80 μm ball. Synaptic connections are shown in gray. *Scale bar* indicates 20 μm. Bottom: Distribution of dendritic lengths. Arrow indicates mean. **b** Example of MF statistics generated with a correlation radius of σ = 20 μm and average fraction of active MFs ($f_{MF}$) of 0.3. Left: Histogram of the fraction of active MFs over different activity patterns. Right: Correlation between MF pairs plotted against distance between them (grey). Black indicates specified $f_{MF}$ (left) or specified spatial correlations (right). **c** Schematic of feedforward network (red, MFs; blue, GCs). The downstream perceptron-based decoder classifies either GC patterns (as shown) or else raw MF patterns without the MF-GC layer. Inset shows the rectified-linear GC transfer function. **d** Example of root-mean-square error as a function of the number of training epochs during learning based on MF (red) or GC (blue) activity patterns. Dashed line indicates threshold error. For this example, $f_{MF} = 0.5$ and the number of inputs per GC ($N_{syn}$) is 4. **e** Raw learning speed of perceptron classifier for different correlation radii, for MFs (red) or GCs with sparse (solid blue, $N_{syn} = 4$) or dense (dashed blue, $N_{syn} = 16$) connectivity. **f** Normalized learning speed (GC speed/MF speed) shown for different synaptic connectivities and fractions of active MFs. Blue lines represent double exponential fit of the boundary at which the normalized speed equals 1 (i.e., when the perceptron learning speed is the same for GC and MF activity patterns). For clarity, only the region in which the normalized speed > 1 is shown. Left: independent MF activity patterns. Right: Correlated MF inputs (σ = 20 μm). **g** Top: Median normalized learning speed (over different $f_{MF}$) for sparse (solid line, $N_{syn} = 4$) and dense (dashed line, $N_{syn} = 16$) synaptic connectivities, plotted against correlation radius. Bottom: Robustness of rapid GC learning for different correlation radii

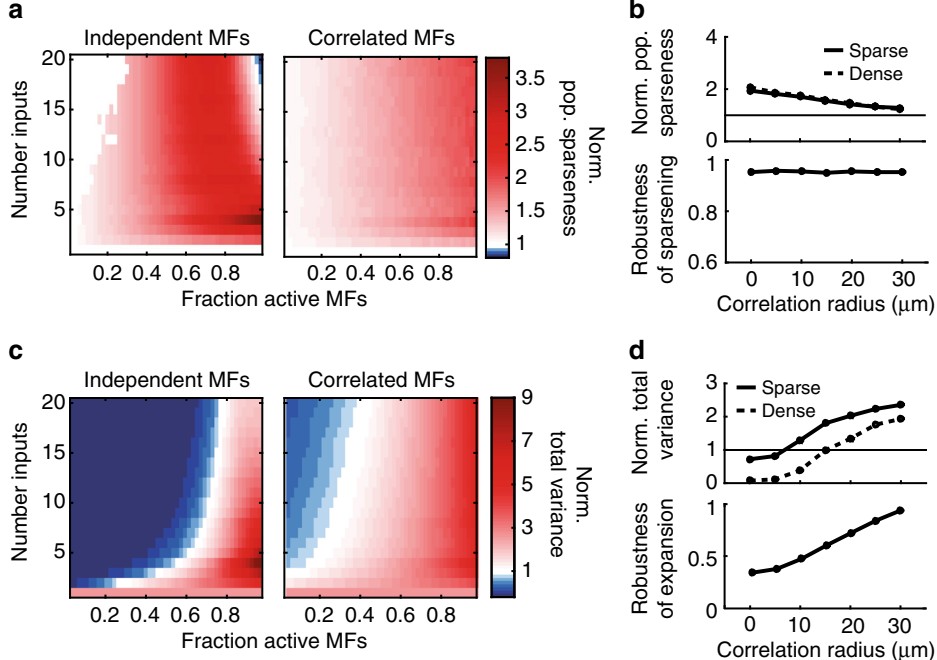

**Fig. 2** Cerebellar input layer sparsens and expands input activity patterns. **a** Normalized population sparseness (granule cell sparseness/mossy fiber sparseness) for independent mossy fiber (MF) activity patterns (left) and correlated MF inputs (right, σ = 20 μm). **b** Top: Median normalized population sparseness for sparse (solid line, $N_{syn} = 4$) and dense (dashed line, $N_{syn} = 16$) synaptic connectivities, plotted against correlation radius. Bottom: Robustness of population sparsening for different correlation radii. **c**, **d** Same as **a**, **b** plotted for normalized total variance

independent mossy fiber firing patterns. But mossy fiber firing patterns can be remarkably diverse, as they encode both discrete[25] and continuous[26, 27] stimuli. Furthermore, their receptive fields are arranged in a large-scale modular structure with a finer 'fractured map' topographical organization[28], that likely results in spatially correlated inputs. How can the relatively homogenous network structure of the cerebellar input layer separate such a diverse range of input activity patterns? We examined the relationship between network structure and pattern separation in the cerebellar input layer by studying how divergent feedforward networks transform highly overlapping, spatially correlated input activity patterns. Using a combination of simplified and biologically detailed models, we disentangled the effects of correlations from expansion and sparsening of spatially correlated input patterns. We quantified pattern separation performance by assaying learning speed using a machine learning algorithm. Our results show that the granular layer is able to perform robust pattern separation over a wide range of mossy fiber firing patterns, but only when the synaptic connectivity of the network is sparse. The performance of divergent feedforward networks was primarily determined by expansion and correlations, rather than sparse coding. Our results establish that the evolutionarily conserved sparse synaptic connectivity found in divergent feedforward networks is essential for separating spatially correlated input patterns.

## Results

**Modeling the cerebellar input layer.** The cerebellar input layer consists of mossy fibers (MFs), which form large en passant mossy-type presynaptic structures called rosettes, granule cells (GCs) which have ~4 short dendrites, and inhibitory Golgi cells which form an extensive dense axonal arbor spanning the local region. To capture the excitatory synaptic connectivity we used an anatomically accurate 3D model of a local region of the GC layer (GCL)[18]. The 80 μm diameter model had experimentally measured densities of MF rosettes ( ~ 180 in total) and GCs ( ~ 480)

and random connectivity, subject to the constraint that MF-GC distances were near 15 μm (Fig. 1a). Importantly, this model reproduced the measured 1:2.9 local expansion ratio between MF rosettes and GCs, the 1:12 divergence at the rosette-GC synapse and the sampling of 4 different rosettes by individual GCs.

To capture spatial correlations in the MF activity patterns, we used a technique to create spike trains with specified firing rates and spike correlations[29]. A Gaussian correlation function was used to describe the distance-dependence of rosette co-activation, which was parameterized by its standard deviation σ (the 'correlation radius'; Fig. 1b). To explore how synaptic connectivity and input correlations affect pattern separation we varied the number of synaptic connections per GC ($N_{syn}$) in the model and presented the networks with different activity patterns while varying the fraction of active MFs ($f_{MF}$) and σ. We implemented a simplified high-thresholding rectified-linear model of GCs and assayed network performance by training a perceptron decoder to classify either MF or GC population activity patterns into randomly assigned classes (Fig. 1c).

**Sparse connectivity speeds learning and increases robustness.** We first tested whether the evolutionarily conserved connectivity in the GCL ($N_{syn} = 4$) could separate MF activity patterns and thus aid learning. Performance was measured by the learning 'speed' of a downstream perceptron decoder (see Methods)[2]. As little is known about which features of GC patterns are relevant for Purkinje cells during learning, we used random classification to assay general pattern separation. Comparison of learning speed when the perceptron was connected to the MF input (red) or the GC output (blue) confirmed that the GCL speeds learning (Fig. 1d). However, network performance depended strongly on input correlations and the density of connectivity (Fig. 1e). Indeed, the learning speed for more densely connected networks ($N_{syn} = 16$, dashed blue line in Fig. 1e) was worse than raw MF input.

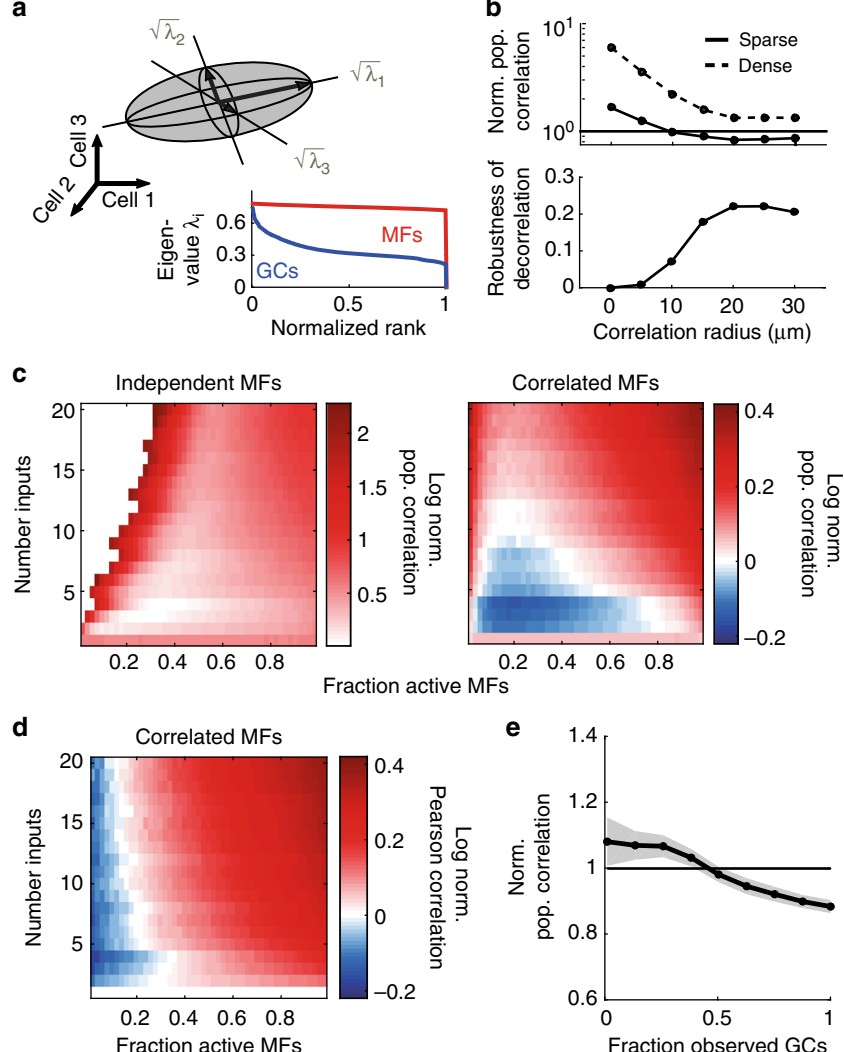

**Fig. 3** Correlations in activity increase with the extent of excitatory synaptic connectivity in feedforward networks. **a** Top: Illustration depicting a distribution of neural activity patterns (grey ellipsoid) in 3D activity space. Mathematically, principal lengths (black arrows) are equal to the square roots of the eigenvalues of the covariance matrix. Bottom: Example of ranked eigenvalues for mossy fiber (MF, red) and granule cell (GC, blue) activity patterns for independent MF inputs. Rank is normalized by dimensionality. Note that the MF eigenvalues are far more uniform than the GC eigenvalues, indicating that the MF patterns are less correlated. In this example, parameters are: $N_{syn} = 4$, $f_{MF} = 0.5$, $\sigma = 0\ \mu m$. **b** Top: Median normalized population correlation (GC correlation/MF correlation) for sparse (solid line, $N_{syn} = 4$) and dense (dashed line, $N_{syn} = 16$) synaptic connectivity plotted against correlation radius. Note the logscale for the population correlation. Bottom: Robustness of GC decorrelation for different correlation radii. **c** Log of the normalized population correlation for independent MF activity patterns (left) and correlated MF inputs (right, $\sigma = 20\ \mu m$). Blue region in the right panel indicates region of active decorrelation of MF patterns (defined by normalized population correlation < 1). **d** Log of the normalized Pearson correlation coefficient for correlated inputs ($\sigma = 20\ \mu m$), averaged over all GC or MF pairs. **e** Average normalized population correlation for subpopulations of increasing size. Grey shading indicates the standard deviation across different samples and observations. For this example, $N_{syn} = 4$ and $\sigma = 20\ \mu m$

To quantify the relationship between synaptic connectivity and learning speed we generated a family of models with different $N_{syn}$ and determined their performance across the full range of $f_{MF}$. To compare network performance across different conditions we normalized the learning speed of the classifier when connected to the GCs by the speed when connected directly to the MFs. For independent MF activity patterns ($\sigma = 0\ \mu m$) the normalized learning speed was substantially increased in networks with few synaptic connections per GC (Fig. 1f, left), especially for high $f_{MF}$. Interestingly, the fastest speed up occurred with ~ 4 synapses per GC. However, as $N_{syn}$ increased, the range of $f_{MF}$ over which the GCL improved learning (i.e., normalized learning speed > 1) decreased.

When spatial correlations were introduced in the MF input, the ranges of $f_{MF}$ and $N_{syn}$ over which the GCL sped learning

increased. However, optimal performance (up to an 8-fold increase) occurred when synaptic connectivity was sparse ($N_{syn} = 2$–5; Fig. 1f, right) and $f_{MF}$ was high, as for the case with spatially independent input. Normalized learning speed increased with $\sigma$ but saturated around 15 μm (Fig. 1g, top). Moreover, the fraction of the parameter space in which GC learning outperformed MF learning (referred to as 'Robustness' of GC learning; Supplementary Methods) also saturated around 15 μm (Fig. 1g, bottom). These results suggest that to improve learning performance in downstream classifiers, cerebellar-like feedforward networks require sparse synaptic connectivity.

**Population sparsening and expansion in coding space.** To understand why sparsely connected feedforward networks

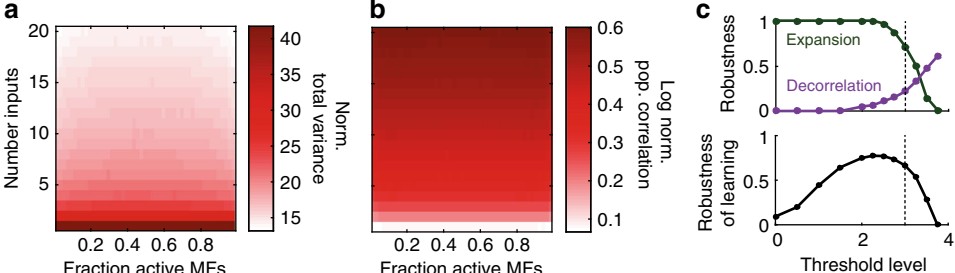

**Fig. 4** Dependence of coding space and correlation on connectivity and the role of thresholding in controlling the expansion and decorrelation. **a** Normalized total variance and **b** log normalized population correlation for networks of linear granule cells (i.e. in the absence of a threshold). Correlation radius is $\sigma = 20$ µm. **c** Top: robustness of expansion (green) and decorrelation (purple) for varying levels of granule cell (GC) threshold. Dotted line indicates the experimentally estimated value of threshold (3 of the 4 mossy fibers, MFs). Bottom: Robustness of learning for varying GC threshold

improve learning, while densely connected networks do not, we analyzed how these networks transform activity patterns. Marr-Albus theory posits that two factors underlie pattern separation in cerebellar cortex: population sparsening and expansion recoding. We first tested whether sparse coding could explain the dependence of learning speed on network connectivity (Fig. 1f) by measuring the population (i.e., spatial) sparseness of GC and MF activity patterns[30]. To compare across parameters we normalized the GC population sparseness by the MF population sparseness. Because of the high GC activation threshold, GC activity was generally sparser than MF activity (Fig. 2a). The normalized population sparseness increased with $f_{MF}$, but was on average similar in magnitude for sparse and dense synaptic connectivities (Fig. 2b, top). Furthermore, increasing $\sigma$ had no effect on the robustness of population sparsening, and actually *decreased* the normalized population sparseness, contrary to the increase expected from the normalized learning speed (cf. Fig. 2b and Fig. 1g, bottom). Therefore the change in normalized population sparseness was unable to account for the effect of network connectivity and MF correlations on learning speed. This suggests that another mechanism (that counters the loss of population sparsening) is responsible for the increase in pattern separation performance for more spatially correlated inputs.

We next considered whether expansion in coding space could explain the trends in pattern separation that we observed. Expansion recoding is thought to speed learning by increasing the distance between patterns in coding space. A key property of such expansion is the size of the distribution of activity patterns, which can be quantified by calculating the total variance in activity of the GC population normalized by the total variance of the MF population (see Methods). The normalized total variance captures both the expansion in dimensionality (due to the 1:2.9 expansion ratio) and any change in the overall size of the population coding space. As the expansion ratio is fixed in our study (except Supplementary Fig. 1), we use the terms "expansion in coding space" and "normalized total variance" interchangeably. Like population sparsening, the normalized total variance increased with $f_{MF}$. However, the normalized total variance better predicted the change in learning speed than the normalized population sparseness (left panels of Figs. 1f and 2c). Still, the total variance tended to underestimate performance of sparsely connected networks and overestimate performance of densely connected ones, particularly for correlated MFs (right panels of Figs. 1f and 2c). Moreover, the magnitude and robustness of the normalized total variance increased approximately linearly with MF correlations (Fig. 2d), unlike the saturation observed for learning speed (Fig. 1g). Qualitatively, this implies that population sparsening

and expansion are not the only factors determining pattern separation performance.

**Decorrelation of MF activity patterns**. We next considered the impact of correlated activity on pattern separation. The presence of spatial correlations in MF inputs is expected to reduce the dimensionality of activity patterns and slow learning due to increased pattern overlap. Mathematically, the shape of the distribution of activity patterns is described by the covariance matrix, since the square roots of its eigenvalues correspond to the lengths of the principal directions of activity space (illustrated in Fig. 3a, top). Independent MF activity results in more uniform eigenvalues (e.g. a sphere in 3 dimensions), whereas more correlated distributions have a more heterogeneous spread of eigenvalues and hence an elongated distribution (Fig. 3a).

To assay neural co-variability we introduced a population-based measure of correlation, calculated using the eigenvalues of the covariance matrix, which captured the elongation of the distribution of activity patterns (see Methods). This "population correlation" varied from 0 for an uncorrelated Gaussian with identical variances (see Supplementary Methods for a discussion on heterogeneous variances) to 1 (e.g., if all neurons have identical activity). Networks with dense synaptic connectivity exhibited considerably higher normalized population correlation (GC population correlation/MF population correlation) than networks with sparse synaptic connectivity irrespective of $\sigma$ (Fig. 3b). This occurred because networks with higher $N_{syn}$ receive a larger number of shared inputs from MFs. In the limit of full connectivity, each GC would be identical, rendering learning impossible. Sparse synaptic connectivity minimizes unwanted GC correlations being introduced by the network structure.

Network structure was not the only factor governing the GC population correlation. Surprisingly, when MF activity patterns were spatially correlated, the population correlation of GCs in sparsely connected networks was often lower than that of the MFs, as revealed by plotting the log of the normalized population correlation (Fig. 3c, right). Such decorrelation of input patterns (normalized population correlation < 1; equivalently, log normalized population correlation < 0) has been shown to arise from thresholding, which attenuates subthreshold input correlations[31]. Contrary to population sparsening and expansion in coding space, the strongest decorrelation occurred for low to intermediate $f_{MF}$. The robustness of pattern decorrelation in our networks saturated when the correlation radius reached $\sigma \sim 15$ µm, potentially explaining the saturation in learning observed previously (Fig. 3b, bottom, cf. Fig. 1g). Moreover, varying the expansion ratio (Supplementary Fig. 1) and including adaptive thresholding to model feedforward inhibition (Supplementary

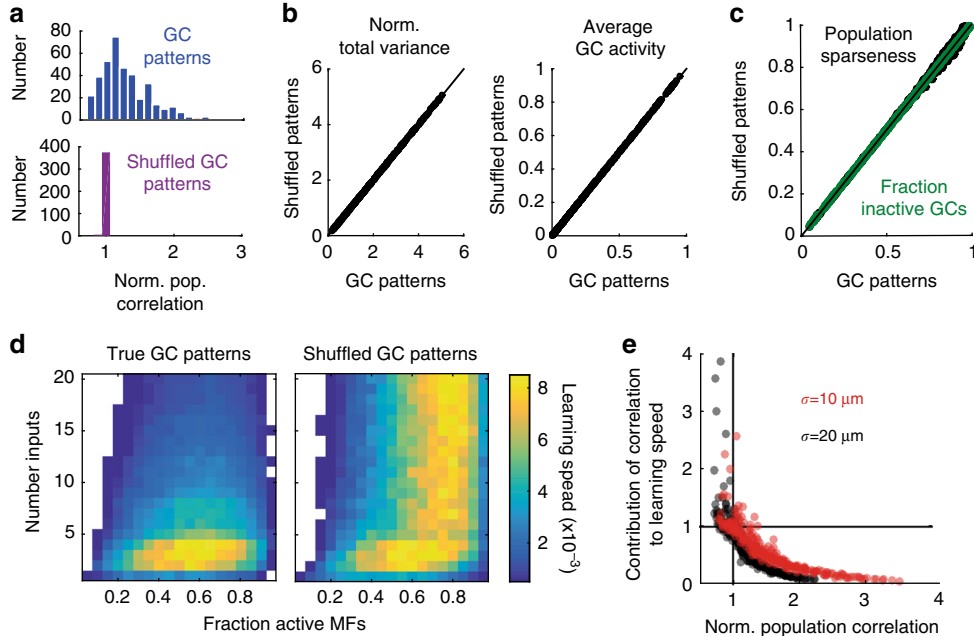

**Fig. 5** Separation of the effects of correlation on learning speed from expansion and population sparsening. **a** Histograms of the normalized population correlation (granule cell correlation/mossy fiber correlation) for granule cell (GC) patterns (top, blue) and shuffled GC patterns (bottom, purple). The narrow distribution around 1 indicates that the shuffled GC patterns have the same normalized population correlation as the mossy fiber (MF) patterns. **b** Normalized total variance (left) and average activity (right) for GC patterns (abscissa) versus shuffled GC patterns (ordinate). **c** Population sparseness (green) plotted for GC patterns (abscissa) and shuffled GC patterns (ordinate). Green indicates the fraction of inactive GCs for comparison as an alternate measure of population sparseness. **d** Raw learning speed for true GC patterns (left) and shuffled GC patterns (right). In both panels, MF inputs are correlated with a correlation radius of $\sigma = 20\,\mu m$. **e** Change in learning speed due to correlations (i.e., GC speed/shuffled GC speed) plotted against the normalized population correlation. Each point represents different values of $N_{syn}$ and $f_{MF}$. Correlation radii were $\sigma = 10\,\mu m$ (red) or $20\,\mu m$ (black)

Fig. 2) produced qualitatively similar results. These results suggest that changes in GC correlations arising from MF input patterns, thresholding, and network structure all play a key role in pattern separation.

When GC correlations were instead assayed with the average Pearson correlation coefficient, rather than population correlation, decorrelation was no longer visible (Fig. 3d). Importantly, the inconsistency between these measurements was not due to insufficient sampling (Supplementary Fig. 3). Instead, this reveals a fundamental property of the decorrelation performed by sparsely connected feedforward networks: the population correlation takes into account the shape of the distribution at the full population-level, while the Pearson correlation only considers the marginal distributions of cell pairs, missing how they may work together to shape the full distribution (see Methods). This has important implications for measuring coordinated activity in these networks, as a large fraction of cells were required to observe decorrelation (e.g. > 50% of the population for strong input correlations; Fig. 3e). Therefore, a substantial proportion of MFs and GCs must be analyzed at the population level in order to accurately measure the extent of decorrelation in the input layer of the cerebellar cortex.

**Determinants of expansion and decorrelation.** To understand how synaptic connectivity and thresholding separately contribute to pattern separation, we next analyzed networks of GCs with linear transfer functions (i.e. in the absence of a threshold), since under these conditions the changes in total variance and population correlation arise solely from network structure. The total variance of linear GCs was larger than that of the MFs over the full range of parameters; however, as $N_{syn}$ increased, the normalized total variance decreased (Fig. 4a) due to GCs averaging the signals across more MFs. Comparison of these

results with those from networks with nonlinear GCs (Fig. 2c, right) shows that thresholding reduces both the magnitude of the expansion of coding space and its robustness (Supplementary Fig. 4). Thus, expansion of coding space is maximal for linear networks ($N_{syn} = 1$), but this is reduced by increasing network connectivity and by GC thresholding.

Linear GC networks also revealed that the network structure introduces considerable population correlation (Fig. 4b). However, this was markedly reduced in networks of nonlinear neurons due to threshold-induced decorrelation (Fig. 3c, right). Previous work has shown that input correlations can be quenched by the presence of intrinsic nonlinearities[31]. Our results show that for feedforward networks, threshold-induced decorrelation of MF input patterns was most pronounced in sparsely connected networks ($N_{syn} \sim 2$–9). Indeed, increasing the threshold increased the region of decorrelation in our networks (Fig. 4c, top; Supplementary Fig. 4), consistent with previous work showing that population sparsening decorrelates inputs[20, 30]. In contrast, the decorrelating effect of thresholding weakened with increasing $N_{syn}$, due to the presence of network-induced correlations in the summed input to each GC. Moreover, decorrelation was not observed for linear networks. Thus GC thresholding enables decorrelation of spatially correlated input patterns only when the synaptic connectivity of the network is sparse and $N_{syn} > 1$.

This reveals a trade-off between expansion of coding space and a reduction of input correlations that depends on both network structure and thresholding. Networks with dense connectivity perform pattern separation poorly because they quench coding space and introduce strong correlations in the output. By contrast, the sparse synaptic connectivity found in many feedforward networks, including the GCL, minimizes output correlations introduced by the network, thereby enabling both expansion of coding space and threshold-induced decorrelation

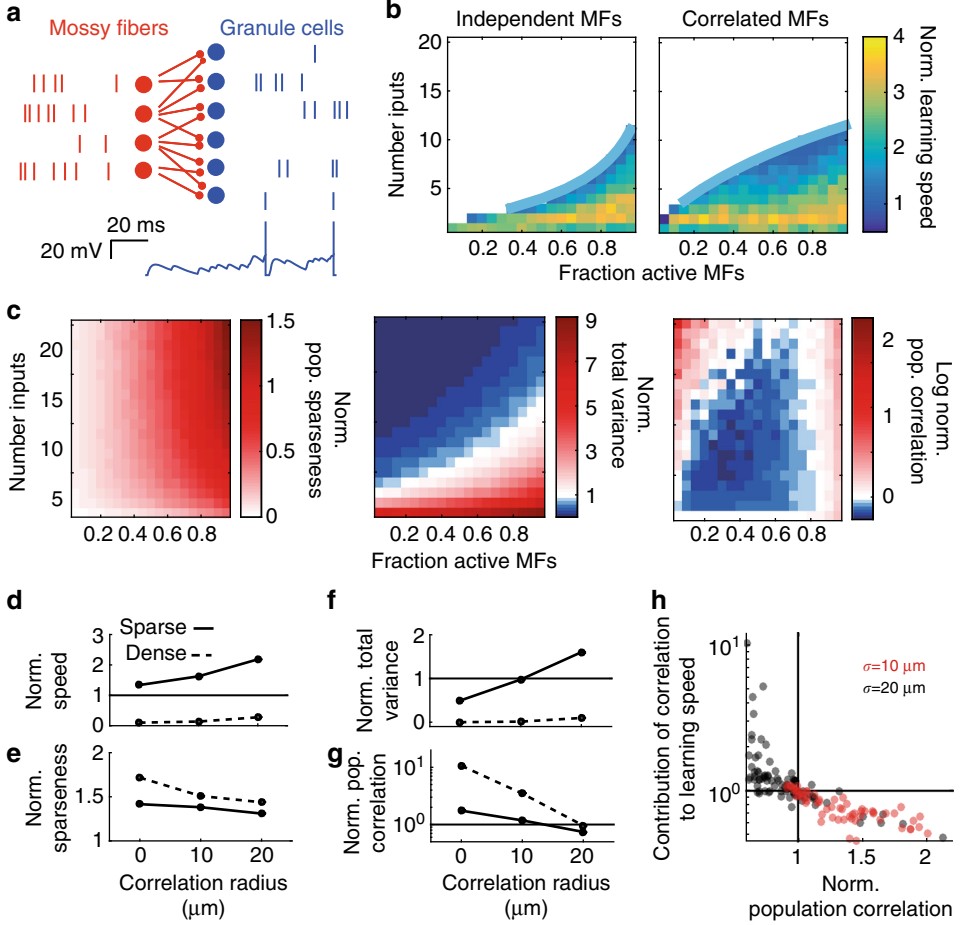

**Fig. 6** Pattern separation and learning speed depend on synaptic connectivity in biologically detailed spiking models of the cerebellar input layer. **a** Top: Schematic of biologically detailed spiking network model with sample spike trains. Bottom: example voltage trace from a granule cell (GC) in network. **b** Normalized learning speed for a spiking network with independent (left) and correlated (right, $\sigma = 20\,\mu m$) mossy fiber (MF) activity patterns. **c** Normalized population sparseness (left), normalized total variance (center), and log normalized population correlation (right) for networks with different numbers of synaptic connections receiving correlated MF activity patterns ($\sigma = 20\,\mu m$). **d** Median normalized learning speed plotted against correlation radius for sparse (solid, $N_{syn} = 4$) and dense (dashed, $N_{syn} = 16$) synaptic connectivities. **e**-**g** Same as **d** for normalized population sparseness **e**, normalized total variance **f**, and normalized population correlation **g**. **h** Change in learning speed due to correlations (i.e., GC speed/speed for shuffled GC spike trains) plotted against the normalized population correlation. Each point represents different values of $N_{syn}$ and $f_{MF}$. Correlation radii were $\sigma = 10\,\mu m$ (red) or $20\,\mu m$ (black)

of input patterns. Moreover, sparsening GC population activity by increasing threshold alters the trade-off between decorrelation and expansion (Fig. 4c, top). This suggests that extremely sparse codes are inefficient for pattern separation and learning due to the quenching of coding space (Fig. 4c, bottom).

**Quantifying the contributions of correlations and decorrelation.** To quantify the contribution of spatial correlations to pattern separation, it was necessary to isolate their effect on learning speed from those arising from population sparsening and expansion of coding space. This required 'clamping' the GC population correlation to the value of the MF population correlation. This constraint necessitated the removal or addition of GC correlations without changing the single-cell statistics (firing rates and variances). To achieve this we extended methods that use random "shuffling" of the timing of activity patterns to remove all correlations[32] by developing an algorithm that shuffled activity patterns to a pre-specified (but nonzero) level of population correlation. The shuffled GC activity distributions had the same population correlation as the MFs (Fig. 5a) while the normalized total variance and firing rates remained unchanged (Fig. 5b).

Importantly, this procedure also maintained the GC population sparseness (Fig. 5c), thereby isolating the effect of correlations from expansion and population sparsening.

Shuffling GC activity patterns to match the MF population correlation had a strong influence on learning speed when compared to the unshuffled control networks, especially for dense synaptic connectivity (Fig. 5d). Unlike the true GC responses, shuffled patterns maintained rapid learning across the full range of $N_{syn}$ examined. These results confirm that the correlations in GC activity induced by network connectivity counteract the positive effects of expansion of coding space, population sparsening, and decorrelation on pattern separation and learning.

We next normalized the GC learning speed by the learning speed using shuffled GC patterns. This enabled us to quantify the effect that GC correlations have on network performance after controlling for expansion and population sparsening. There was a strong negative correlation between the normalized population correlation and learning (Fig. 5e), showing that population correlation reduces the normalized learning speed to as low as 0.05 (corresponding to a 20-fold reduction). In contrast, learning speed was enhanced (up to a 4-fold increase; see Fig. 5e) in sparsely connected networks where the relatively weak network-

dependent correlations in the summed inputs were quenched by threshold-mediated decorrelation. Thus in networks with sparse synaptic connectivity, expansion of coding space and active decorrelation combined for faster, more robust pattern separation and learning.

**Pattern separation performed by a detailed spiking model.** To test the validity of the predictions from our noise-free simplified models, we performed simulations with biologically detailed spiking models of the GCL (Fig. 6a). MFs were modeled as rate coded Poisson spike trains as observed in vivo[25–27] and GC integration was modeled with integrate-and-fire dynamics with experimentally determined input resistance and capacitance, as well as AMPA and NMDA receptor-type excitatory synaptic conductances that included spillover components and short-term plasticity[18]. The tonic $GABA_A$ receptor-mediated inhibitory conductance present in GCs was also included[33]. This level of description reproduces the measured GC input-output relationship[34, 35]. The synaptic connectivity of the detailed model was identical to the rectified-linear model. A downstream decoder was trained to classify MF, GC, or shuffled GC spike counts in a 30 ms window, corresponding to the effective integration time of GCs[18, 35]. Despite the stochastic noise introduced by the Poisson input trains, networks with the sparse level of synaptic connectivity found in the GCL sped learning by up to 4-fold. Detailed models also exhibited the same general trends for pattern separation and learning that were present in the simplified model: learning was fastest for sparsely connected networks, while densely connected networks performed worse than MFs (Fig. 6b). Moreover, the robustness of the normalized learning speed increased with input correlations for sparsely connected networks, but did not significantly increase for densely connected networks.

To examine how population sparsening and expansion of coding space contributed to the speed up in learning in detailed spiking models we first examined the normalized population sparseness of the spike count patterns. The increase in the normalized population sparseness with the number of synaptic inputs was more pronounced than for the simplified model (Fig. 6c, left). This is likely caused by the fact that the GC input-output nonlinearity sharpens as $N_{syn}$ increases, as shown by previous modeling[18]. However, while the normalized learning speed increased with input correlations in sparsely connected networks (Fig. 6d), the normalized population sparseness decreased (Fig. 6e). Therefore, sparse encoding could not explain the dependence of learning on MF correlations. In contrast, the normalized total variance had a similar dependence on $N_{syn}$ and $f_{MF}$ as the normalized learning speed (Fig. 6c, center). Moreover, like the normalized learning speed, the normalized total variance in sparsely connected networks increased with MF correlations, while densely connected networks exhibited little change (Fig. 6f). However, the normalized total variance did not capture the full magnitude of the speedup for sparsely connected networks. Interestingly, decorrelation was more robust in the detailed spiking model than for the simplified model (Fig. 6c, right). Like the normalized population sparseness, this likely arises from the change in the nonlinearity of the GC input-output relationship with increasing $N_{syn}$. In line with predictions from our simplified model, as $\sigma$ increased, the normalized population correlation decreased (Fig. 6g). Finally, upon shuffling GC spike count patterns, we found a strong negative relationship between the population correlation and its impact on learning, with decorrelation speeding learning beyond the effects of expansion, as predicted by our simplified model (Fig. 6h c.f. Fig. 5e). These results show that the network connectivity and biophysical

mechanisms present in the GCL can implement effective pattern separation in the presence of noise. Moreover they confirm the predictions from our simplified models, which show that sparse connectivity and nonlinear thresholding is essential for effective pattern separation and decorrelation in feedforward excitatory networks.

## Discussion

We have explored the relationship between the structure of excitatory feedforward networks and their ability to perform pattern separation. To do this we examined how simplified and biologically detailed network models with varying synaptic connectivity transform spatially correlated activity patterns, and how this transformation affects the learning speed of a downstream classifier. Our results reveal that the structure of divergent feedforward networks governs pattern separation performance because increasing synaptic connectivity increases correlations in the output, counteracting the beneficial effects of expansion of coding space. Moreover, only in networks with few synaptic connections per neuron, as found in the cerebellar GCL, can spike thresholding actively decorrelate input activity patterns. The pattern separation performance of sparsely connected networks was robust to a wide range of MF statistics and to both sparse and dense regimes of GC firing. Our work suggests that sparse synaptic connectivity is essential for separating spatially correlated input patterns and enabling faster learning in downstream circuits.

The idea that divergent feedforward networks separate overlapping patterns by expanding them into a high-dimensional space has a long history. In the cerebellum, pioneering work by Marr and Albus linked the structure of the GCL to expansion recoding of activity patterns[1, 2]. Subsequent theoretical work has broadened our understanding of how pattern separation, information transfer, and learning arise in cerebellar-like feedforward networks[3, 6, 7, 18, 19, 36, 37]. Our work extends these findings in several ways. First, we gained new insight into pattern separation by isolating the effects of input decorrelation, expansion of coding space, and population sparsening. While these mechanisms have been identified previously as factors supporting pattern learning in cerebellar-like systems, the contribution of each factor has not been clear. Through our analyses, we identified expansion and decorrelation, rather than sparse coding, as the key mechanisms underlying pattern separation. Second, previous work analyzed idealized, uncorrelated input patterns, raising the question of whether efficient pattern separation extends to more realistic inputs. We investigated MF patterns with a wide range of activity levels and spatial correlations, finding that the performance of sparsely connected networks is robust to diverse input properties. Finally, we showed that biologically detailed spiking models with the sparse synaptic connectivity present in the GCL can decorrelate spatially correlated synaptic inputs, perform pattern separation, and speed learning by a downstream classifier.

Classical studies have highlighted the importance of sparse coding for pattern separation[1, 2, 4, 7, 37]. Moreover, our previous work showed that the sparse synaptic connectivity in the GCL is well suited for performing lossless sparse encoding[18]. However, recent in vivo imaging suggests that GC population activity is denser (50–66% of GCs active) than previously believed[38, 39]. Although these population coding levels were determined over longer timescales than the physiologically relevant GC integration window (due to the slow kinetics of genetically encoded indicator GCaMP6), their high levels potentially cast doubt on Marr-Albus theory of pattern separation in the cerebellum. Our findings show that sparse coding is less important for pattern separation than previously thought. Indeed, the GCL could improve learning both

in sparse and in dense regimes of GC activity (Supplementary Fig. 5), and did not require the extreme sparse coding regimes (i.e., < 5% of GCs active) envisioned by Marr and Albus[1, 2]. In fact, excessive population sparsening quenches the coding space (Fig. 4c), resulting in a loss of information[18]. Thus, we found that, while population sparsening contributes to pattern separation[4, 6, 7, 37], the main determinants are expansion of coding space and correlations.

MFs arise from multiple precerebellar nuclei in the brainstem and often project to specific regions in the cerebellar cortex, resulting in a large-scale modular structure[40]. Within an individual module, the MF receptive fields form a 'fractured map'[28], which is likely to lead to spatially correlated activity at the local level. While single GC recordings suggest multimodal integration[41], in forelimb regions synaptic inputs can convey highly related information[42, 43]. Moreover, because MFs encode both discrete and continuous sensory variables, some cerebellar regions, such as the whisker system in Crus I/II, are likely to experience bouts of intense high frequency MF excitatory drive (100–1000 Hz) interspersed by quiescence[25, 44], while others (e.g., vestibular and limb areas) may experience more slowly modulated input (10–100 Hz)[26, 27]. Our findings suggest that the same optimal network structure can perform decorrelation and pattern separation for a wide range of MF correlations and excitatory drive. Expansion of the coding space and population sparsening were strongest for independent input patterns with many active MFs, while active decorrelation boosted learning for spatially correlated input patterns with fewer active MFs. Although there are likely to be region-specific specializations in synaptic properties and inhibition, the uniformity of GCL structure suggests that it acts as a generic preprocessing unit that decorrelates and separates dense MF activity patterns, enabling faster associative learning in the molecular layer.

Inhibition has been shown to sparsen and decorrelate neural activity patterns[11, 18, 45–47]. Inhibition in the GCL consists of a large fixed tonic $GABA_A$ receptor-mediated inhibition of GCs that is complemented by a weaker activity-dependent component mediated by phasic release and GABA spillover from Golgi cells[33, 48–50]. When network-activity dependent thresholding was included to approximate feedforward Golgi cell inhibition of GCs[18, 51], we observed greater decorrelation (Supplementary Fig. 2) because the increasing threshold filters out a substantial proportion of the correlated input. However, the qualitative dependence of pattern separation on network connectivity was preserved.

Because pattern separation is essential for a wide range of sensorimotor processing, it is not surprising that divergent feedforward excitatory networks are found throughout the brain of both vertebrates and invertebrates. Interestingly, the synaptic connectivity in many of these networks is sparse[16, 17, 52]. Furthermore, the characteristic 2-7 synaptic connections found in the GCL has been evolutionarily conserved since the appearance of fish[53]. Our results indicate that such sparse connectivity is optimized for decorrelation and pattern separation, regardless of the precise expansion ratio (Supplementary Fig. 1). These results agree with recent analytical modeling, which predicts that the levels of sparse connectivity observed for GCs (and Kenyon cells in fly) are optimal for learning associations[19]. This suggests that the advantage of improved pattern separation and learning that sparse synaptic connectivity confers has been sufficient to conserve the structure of the GCL for 300–400 million years.

A core function of the cerebellar cortex is to learn the sensory consequences of motor actions, allowing it to refine motor action and to enable sensory processing during active movement[54–56]. In Purkinje cells, learning is achieved by altering synaptic strength depending on the timing between GC activity and feedback error via climbing fiber input[57]. We used perceptron-based learning to assay pattern separation performance, since theoretical work has recognized analogies between supervised learning in Purkinje cells and perceptrons[2, 58, 59]. However, important functional differences with Purkinje cells limit finer-grained insights into cerebellar learning. Moreover, we tested random pattern learning because it is a general and challenging task. Once more is known about which features of GC activity are relevant for motor learning, it will be interesting to see whether structured connectivity makes expansion recoding more effective by reducing the variance between functionally similar activity patterns[6]. Regardless of the precise classification task, our results reveal the essential role sparse synaptic connectivity plays in minimizing correlations. It will also be interesting to investigate whether sparse synaptic connectivity confers comparable improvements in temporal pattern learning, since temporal expansion will increase the dimensionality of the system further[10, 60–62].

Our results are consistent with several existing experimental manipulations in the cerebellar cortex. Reducing the number of functional GCs by 90% using a genetic manipulation that blocked their output resulted in deficits in the consolidation of motor learning[63]. Our findings suggest that this phenotype arose from the reduced coding space. Another prediction is that decreasing GC threshold will affect the expansion-correlation tradeoff, reducing pattern separation performance. Interestingly, lowering the spike threshold by specifically deleting the KCC2 chloride transporter in cerebellar GCs resulted in impaired learning consolidation[64]. Similarly, inhibiting a negative feedback circuit in the drosophila olfactory system increased correlations in odor-evoked activity patterns and impaired odor discrimination[11]. These findings are consistent with our prediction that lowering threshold increases output correlations in feedforward networks and impairs pattern separation and learning.

The most direct experimental test of this work is to compare the total variance and population correlation of GC and MF spiking patterns. However, our results show that pairwise correlations may not capture active decorrelation, consistent with previous work that showed pairwise measurements can underestimate collective population activity[65]. Our analysis indicates that dense recordings from a large fraction of the neurons in the local network are required to measure population correlation in MFs and GCs (Fig. 3e). Recent developments in high speed random access 3D two-photon imaging[66, 67] and genetically encoded $Ca^{2+}$ indicators[68] potentially make this type of challenging measurement feasible for the first time. Application of these new technologies would provide direct experimental tests of our findings, thereby improving our understanding of how spatially correlated activity patterns are transformed and separated in the cerebellar cortex.

## Methods

**Anatomical network model**. Both the simplified and biophysical models used an experimentally constrained anatomically realistic network connectivity model of an 80 μm diameter ball within the granular layer[18]. MF rosettes and GCs were positioned according to their observed densities. GCs were connected to a fixed number ($N_{syn}$) of MFs, which were chosen randomly while constraining the MF-GC distance to be as close as possible to 15 μm, the average dendritic length.

**Spatially correlated input patterns**. MF activity patterns were created using a method based on Dichotomized Gaussian models that generates binary vectors with specified average values and correlations[29]. The average value of the binary vector represented the fraction of active MFs ($f_{MF}$). The correlation coefficient between two MF patterns was chosen to be a Gaussian function of distance with the correlation radius parameterized by its standard deviation σ. For the simplified model, these binary patterns were used directly. For the detailed model, activated MFs fired at 50 Hz while inactivated MFs were silent. Note that this method is distinct from recent papers studying patterns that are arranged into clusters in *state*

space representing e.g. specific odorants[6, 19], as they lack the correlated structure in *physical* space that impede learning (Supplementary Fig. 6).

**Simplified network model**. GC activity was given by:

$$x_i^{GC} = f^+\left(\sum_j \frac{4}{N_{syn}} C_{ij} x_j^{MF} - \theta\right)$$

where $N_{syn}$ is the number of synaptic inputs per GC, $C_{ij}$ is the binary connectivity matrix determined by the anatomical network model, and $f^+$ is a rectified-linear function, i.e., $f^+(x) = \max(0,x)$. Unless otherwise specified, the threshold was set to $\theta = 3$, in line with experimental evidence that three MFs on average are required to generate a spike in GC[34, 69].

**Biologically detailed network model**. MFs were modeled as modified Poisson processes with a 2 ms refractory period and firing rate determined by the generated binary activity patterns described above (50 Hz if the MF was activated, silent otherwise). GCs were based on a previously published model of integrate-and-fire neurons with experimentally measured passive properties and experimentally constrained AMPA and NMDA conductances, short-term plasticity and spillover components as well as constant GABA conductance representing tonic inhibition[18] (see Supplementary Methods). The model was written in NeuroML2 and simulated in jLEMS[70]. For learning and population-level analysis, activity patterns were defined as the vector of spike counts in a 30 ms window (after discarding an initial 150 ms period to reach steady state).

**Implementation of perceptron learning**. A perceptron decoder was trained to classify 640 input patterns into 10 random classes. Random classification was chosen to ensure maximal overlap between patterns. The number of classes was chosen to be slightly under the memory capacity for a wide range of parameters, allowing comparison of learning in different networks for a relatively complex task. Online learning was implemented with backpropagation learning on a single layer neural network with sigmoidal nodes and a small fixed learning rate of 0.01. The inputs consisted of either the raw MF or the GC activity patterns. Learning took place over 5000 epochs, each of which consisted of presentations of all 640 patterns in a random order. Learning speed was defined as $1/N_E$, where $N_E$ is the number of training epochs until the root-mean-square error reached a threshold of 0.2. Other error thresholds gave qualitatively similar results.

**Analysis of activity patterns**. Population sparseness was measured as[30]:

$$\frac{N - \frac{\left(\sum_i x_i\right)^2}{\sum_i x_i^2}}{N-1}$$

where N is the number of neurons and $x_i$ is the ith neuron's activity (simplified model) or spike count (detailed model). The above quantity was averaged over all activity patterns. To quantify expansion of coding space, we use the total variance, i.e. the sum of all variances:

$$\sum_i \text{var}(x_i)$$

We defined the population correlation as:

$$\frac{N}{N-1}\left(\frac{\max\{\sqrt{\lambda_i}\}}{\sum_i \sqrt{\lambda_i}} - \frac{1}{N}\right)$$

where $\lambda_i$ are the eigenvalues of the covariance matrix of the activity patterns. The first term in this expression describes how elongated the distribution is in its principal direction. The second term subtracts the value 1/N so that an uncorrelated homogenous Gaussian would have a value of zero. A modified version of the population correlation to control for heterogeneous variances did not affect the results (see Supplementary Methods). Finally, the scaling factor of $\frac{N}{N-1}$ normalizes the expression so that its maximum value is 1. Both the population correlation and the correlation coefficient describe covariability between pairs of cells. However, the population correlation contains additional information about how those pairs constrain the shape of the full distribution.

**Partial shuffling of spiking activity**. We developed a shuffling technique to increase or decrease the population correlation to a desired level, while keeping the mean and variance of each neuron fixed. First, to shuffle GC patterns to a lower level of correlation, for each neuron we took two random GC patterns and exchanged the value of that neuron's spike count in one pattern with its spike count in the other pattern. This step was iterated over the full population and over random pattern pairs until the resulting activity patterns had the desired population correlation. Conversely, to shuffle activity patterns in a way that would increase correlations, we took random pairs of patterns and swapped the activity so

that each cell had a lower spike count for the first pattern and higher activity for the second pattern. This procedure modifies the activity patterns so that the population overall tends to be more active together. We then tested perceptron learning based on the new shuffled activity patterns. See Supplementary Methods for additional details.

**Data availability**. Models and scripts for running and analyzing simulations are available at https://github.com/SilverLabUCL/MF-GC-network-backprop-public. All scripts necessary for simulation data are included, as well as pre-simulated data from the biologically detailed spiking model necessary to reproduce Fig. 6 (see above).

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

## Acknowledgements

This work was supported by the Wellcome Trust (095667; 203048; 101445 to R.A.S. and 200790 to C.C.). R.A.S. is in receipt of a Wellcome Trust Principal Research Fellowship and an ERC advanced grant (294667). We thank Eugenio Piasini, Sadra Sadeh, Yann Sweeney, Antoine Valera and Tommy Younts for comments on the manuscript and Ashok Litwin-Kumar and Kam Harris for helpful discussions.

## Author contributions

N.A.C.-G. carried out the simulations and analyzed the data. N.A.C.-G., C.C and R.A.S. conceived the project and designed the experiments. N.A.C.-G and R.A.S. wrote the manuscript.

## Additional information

**Competing interests:** The authors declare no competing financial interests.

