## [Peer Review File · Nature Communications]

Reviewers' comments:

Reviewer #1 (Remarks to the Author):

Cayco-Gajic et al have used computational modeling to investigate the key features of divergent networks that aid pattern separation. For random, spatially correlated inputs, they find that sparse connectivity resembling that in biological networks is near optimal, and demonstrate that the advantages of sparse connectivity come mainly from expansion recoding and decorrelation rather than from sparse coding per se. They have focused primarily on the input layer of the cerebellar cortex but their results have important implications for similar networks found across species. The work is interesting and of high quality. There are some issues with presentation and particularly clarity that should be addressed.

1. The authors have done a nice job dissociating several features of divergent networks to identify which ones contribute most meaningfully to pattern separation. In general, however, multiple dimensions are simultaneously investigated, leading to multiple 2-dimensional plots per figure that are then compared against each other, within and across figures. For this reviewer, it would have been helpful to ground the study with biological constraints earlier on, to motivate the dissociation of the various features of divergent networks. For instance, the need to decorrelate spatially correlated inputs could be introduced in the Introduction – this justification only comes in the Discussion, despite the fact that spatial correlations are a key motivation already from the first figure. The statement at the bottom of p. 22, “our results highlight the detrimental effect that spatial correlations can have on sensorimotor learning and the substantial improvements in performance that can be achieved through spatial decorrelation and expansion of coding space,” is very clear and language like this in the abstract/ intro would be helpful. As an example, in the abstract, rather than “the relationship between the synaptic connectivity within these circuits and their ability to separate patterns is poorly understood”, a more concrete question could be framed: “how do spatially correlated inputs impact sensorimotor learning and what key features allow divergent networks to improve performance?” (or similar). This kind of approach to the motivation for the study could more gently guide the reader through the network dissection to come.

2. The authors' conclusions about optimal features of divergent networks are based on the assumptions of random, spatially correlated inputs. The dependence on spatial correlations is thoroughly investigated. How do the results depend on the assumption of randomness? An explicit discussion of the assumptions of random, spatially correlated inputs and their implications could follow the paragraph that ends on the bottom of p. 18.

3. Another interesting biological constraint that is investigated in the models but not discussed in the motivation for the study is the activity range of mossy fiber inputs. The authors point out in the Discussion(p 21) that mossy fibers can fire with a large range of temporal activation patterns both in terms of firing frequency and in transient vs sustained signals. Many of the plots have MF activation level on the x-axis, but I did not readily find a discussion of how this parameter was defined or calculated. I may have missed it, but given that some of the results hold for specific ranges of MF activity, I think this issue merits treatment within the Results section itself. It could also be interesting to discuss that the results suggest that sparse connectivity improves learning more at higher MF activity levels (Fig 1), while decorrelation in sparse networks is particularly effective with *low * MF activation (Fig 3).

4. The authors argue (bottom of p. 8, Fig 2 vs. Fig 1) that learning speed “cannot be accounted for by “changes in the sparseness and the size of the coding space alone.” This is, as the authors point out, assuming a linear relationship between each of these features and learning speed, and assuming their

independence. The basis and importance of these assumptions should be discussed.

5. Additional points on clarity:

- Abstract last two sentences unclear – “governing performance by counteracting beneficial effects” is awkward phrasing and it leads to confusion as to what “this” in “this tradeoff” refers to.
- Please use consistent terminology for individual model parameters across methods, results, figures, and legends. In particular, the authors could be more clear on when they are referring to sparse connectivity vs sparse coding (be aware that “sparsening”, “sparseness” etc in this context can be ambiguous). Also please specify when referring to input layer correlations vs output/ gc layer correlations.

Reviewer #2 (Remarks to the Author):

This paper is a very thorough computational study of the expansion recoding idea in the context of feedforward circuitry, originally proposed by Marr and elaborated by Albus. There have been a number of papers over the past decades simulating cerebellar networks, including by this submission’s authors. Now this submission is likely the most exhaustive study yet of the specific problem of pattern separation in a feedforward network. While the results are based on the mossy fiber (MF) to granule cell (GC) synapse in cerebellum, and its modulation by Golgi cell inhibitory input, they are applicable to feedforward nets generally, where one suspects an expansion of the coding space in the presence of divergent connectivity.

Previous work in Silver’s group has shown lossless sparse encoding by sparse synaptic connectivity. This submission provides further insight into how the putative pattern separation works in terms of the tradeoff between sparseness and decorrelation caused by expansion of dimensionality and coding space (assessed with a measure of the variance of the activity patterns), and the nonlinear thresholding by the neurons. It does so by focusing on the question of why learning speed varies with the level of correlation between MF input patterns.

Separation is assessed by the speed of learning of a simple perceptron decoder at the output of the granule cells - which of course is a first step in cerebellar processing (this would then feed into the Purkinje fiber network, but that is not studied here). The results are obtained for a simple threshold linear model, as well as for a more biophysically detailed model.

The authors show that the increase in population sparseness going across the MF-GC synapse can not explain the learning speed’s dependence on input cross-correlation; rather, sparseness only aids pattern separation for independent inputs. For the more realistic correlated input pattern case, expansion of coding space using a large number of GCs and decorrelation caused by thresholding (i.e. nonlinearity that filters out subthreshold correlations) explain the behaviour of learning speed. This is the main result. The model also shows that when network activity further affects thresholds - to approximate real feedforward inhibition from MF to Golgi cells - the decorrelation improves further still.

Most techniques used here are standard; of note is the novel way they randomly shuffle activity patterns all the while preserving population correlations, total variance and mean activity. The strength of the paper is really in bringing a lot of ideas together in a confined computational context, and disentangling certain effects behind expansion coding.

This computational study is thoroughly done and beautifully presented. It is very solid work. The

supplementary material even looks at whether the story changes if other expansion ratios from mossy to granule change, since the glomerulus synapse is a very complex system and the ratio is not clear - this also allows them to generalize their findings to other feedforward systems with different ratios. The SI also looks at adaptive thresholding.

In spite of this praise for the work, in the end I am left with the uneasy recognition that, in spite of all this analysis, there are no real surprises and fundamental new revelations, and the sole focus on explaining how learning speed depends on input correlation is slightly narrow. Partial effects on which the story is built here have previously been described in the literature, such as the need to observe a large fraction of cells to say anything statistically significant about decorrelation. The recent Babadi-Sompolinsky paper, although cited, receives little further attention in the Discussion despite its focus on similar issues with clustered inputs. The current submission interestingly provides a rationale for the observed 4 synapses per GC cell observed, as it led to the highest learning speed. Yet it is somewhat obvious that there has to be a moderate connectivity ratio, since connecting 1 to 1 makes the granule cells linear and incapable of decorrelating with thresholding, and a large ratio leads to too much correlation that is difficult to remove, especially for correlated inputs.

The main finding that sparseness is a weaker determinant of the learning speed vs input correlation does provide "insight", as the authors write, but in my mind does not constitute a sufficiently significant advance or breakthrough that is more required for NCOMM. While it is close to the bar, I consider the work more suitable for other general neuroscience journals Scientific Reports, J. Neuroscience, eLife and the like.

Specific minor points:

p.7 When you mention that you first tested whether sparse coding could explain the dependence of learning speed on connectivity, remind the reader that this addresses Fig.1f.

top of p.8: the sparseness effect actually goes in the opposite direction. So if I understand correctly, one should mention that, in fact, whatever does end up explaining the increase in normalized learning speed with input cross-correlation has to also work against this loss of sparseness.

p.8 bottom: "...and the size of the coding space": I assume you meant "variance" instead of "size" here? You should make it clear whether you consider these terms interchangeable or not.

p.10: when you mention Pearson coefficient, explain it more: emphasize that it is a pairwise measure, averaged over all (?) pairs, and still deals with a normalized GC/MF measure?

p.13: Fig.5d: why not call this the number of inputs as in previous plots? (same comment for the ordinate of Fig.3c,d).

p.14 "In contrast, learning was enhanced (up to a 4-fold increase)...": where is this shown?

p.16: "...this is likely due to the change in the GC input-output function..." : this is the second time one reads this reason in less than a page, and it refers to previous work by the authors. The reader needs to know what this change is to properly understand what is meant.

p.22: "Nevertheless, decorrelation of GC activity is expected to improve the performance of the conversion of spatial activity into temporal sequences via these mechanisms." This expectation needs to be clarified, it seems a big leap.

p.25 top: typo: diameter ball of within the granular layer

p.27: specify: correlation matrix of what ?

p.28 top: "...we swapped the spike counts for random pairs of patterns." : not clear what this means.

In the figures and text, MF activity is never clearly defined. I assume it is, as usual, the fraction of active MF cells.

Fig.1b: tick marks are missing

Fig.3: specify whether you equate active decorrelation with negative normalized correlation.

Replies to the reviewers' comments

We thank the reviewers and the editor for their constructive comments. We have revised the manuscript in line with the suggestions. This has strengthened our study and better highlighted the novel aspects of our findings.

Reviewer 1

Cayco-Gajic et al have used computational modeling to investigate the key features of divergent networks that aid pattern separation. For random, spatially correlated inputs, they find that sparse connectivity resembling that in biological networks is near optimal, and demonstrate that the advantages of sparse connectivity come mainly from expansion recoding and decorrelation rather than from sparse coding per se. They have focused primarily on the input layer of the cerebellar cortex but their results have important implications for similar networks found across species. The work is interesting and of high quality. There are some issues with presentation and particularly clarity that should be addressed.

We have modified the Abstract, Introduction, and Discussion to use more straightforward language and improve clarity. We have also ensured that the terminology is now consistent throughout the manuscript.

1. The authors have done a nice job dissociating several features of divergent networks to identify which ones contribute most meaningfully to pattern separation. In general, however, multiple dimensions are simultaneously investigated, leading to multiple 2-dimensional plots per figure that are then compared against each other, within and across figures. For this reviewer, it would have been helpful to ground the study with biological constraints earlier on, to motivate the dissociation of the various features of divergent networks. For instance, the need to decorrelate spatially correlated inputs could be introduced in the Introduction – this justification only comes in the Discussion, despite the fact that spatial correlations are a key motivation already from the first figure.

We now set up the need to separate spatially correlated inputs in the Abstract and Introduction (p.4), as suggested.

The statement at the bottom of p. 22, “our results highlight the detrimental effect that spatial correlations can have on sensorimotor learning and the substantial improvements in performance that can be achieved through spatial decorrelation and expansion of coding space,” is very clear and language like this in the abstract/ intro would be helpful. As an example, in the abstract, rather than “the relationship between the synaptic connectivity within these circuits and their ability to separate patterns is poorly understood”, a

more concrete question could be framed: “how do spatially correlated inputs impact sensorimotor learning and what key features allow divergent networks to improve performance?” (or similar). This kind of approach to the motivation for the study could more gently guide the reader through the network dissection to come.

We have reworded the Abstract to use clearer language and to motivate the question from a more biological perspective. We have also altered the Introduction (p.4) to address these aspects so that the language is more concrete.

2. The authors’ conclusions about optimal features of divergent networks are based on the assumptions of random, spatially correlated inputs. The dependence on spatial correlations is thoroughly investigated. How do the results depend on the assumption of randomness? An explicit discussion of the assumptions of random, spatially correlated inputs and their implications could follow the paragraph that ends on the bottom of p. 18.

In the absence of knowledge about which features of GC population activity are functionally relevant for Purkinje cells during motor learning, we used random classification as an assay of pattern separation performance, because it is a general and challenging pattern separation problem. Once more is known about associative learning, it will be interesting to see whether structured connectivity makes expansion recoding more effective by reducing the variance between activity patterns that are functionally similar from the Purkinje cell’s point of view, thereby further increasing the separation between functionally distinct patterns. Regardless of the precise classification task, our results reveal the essential role sparse synaptic connectivity plays in minimizing correlations and enabling spatial decorrelation of inputs. These points are now raised in the Discussion (p.23) and the assumption of random inputs for learning is mentioned in the Results (p.6).

3. Another interesting biological constraint that is investigated in the models but not discussed in the motivation for the study is the activity range of mossy fiber inputs. The authors point out in the Discussion(p 21) that mossy fibers can fire with a large range of temporal activation patterns both in terms of firing frequency and in transient vs sustained signals.

We now discuss the diversity of MF input properties and the implications for pattern separation as motivation in the Introduction (p.4), as suggested.

Many of the plots have MF activation level on the x-axis, but I did not readily find a discussion of how this parameter was defined or calculated.

We agree that the MF activation level was not described clearly in the original text. We now define it more explicitly in the Results (p.6) as the “fraction of active MFs”, or f_{MF} , as suggested by Reviewer 2.

*I may have missed it, but given that some of the results hold for specific ranges of MF activity, I think this issue merits treatment within the Results section itself. It could also be interesting to discuss that the results suggest that sparse connectivity improves learning more at higher MF activity levels (Fig 1), while decorrelation in sparse networks is particularly effective with *low * MF activation (Fig 3).*

As mentioned by the reviewer, the largest increase in learning speed occurred when the fraction of active MFs (f_{MF}) was high. Expansion of the coding space and population sparsening were strongest for independent input patterns with many active MFs, while active decorrelation boosted learning for spatially correlated input patterns with fewer active MFs (when inputs were spatially correlated). We have now commented on the determinants of pattern separation at different f_{MF} in the Results (p. 7-9,11) and in the Discussion (p. 21).

4. The authors argue (bottom of p. 8, Fig 2 vs. Fig 1) that learning speed “cannot be accounted for by “changes in the sparseness and the size of the coding space alone.” This is, as the authors point out, assuming a linear relationship between each of these features and learning speed, and assuming their independence. The basis and importance of these assumptions should be discussed.

The perceptron is a linear classifier and it is therefore more likely for the learning speed be linear in sparseness and expansion than if we had used a nonlinear classifier such as k-nearest neighbours. Moreover, the initial comment that the learning speed could not be accounted for by sparseness and expansion, assuming linearity, was a qualitative justification to motivate the investigation of correlations. We treat this rigorously later on when we use a shuffling algorithm to precisely quantify the effect that sparseness and expansion have on learning, independent of correlations.

We have modified the text to make it clear that the qualitative difference between normalized population sparseness, normalized total variance, and normalized learning speed is a motivation for investigating correlations (p.10).

5. Additional points on clarity:

- Abstract last two sentences unclear – “governing performance by counteracting beneficial effects” is awkward phrasing and it leads to confusion as to what “this” in “this tradeoff” refers to.

We have removed these sentences from the abstract.

- Please use consistent terminology for individual model parameters across methods, results, figures, and legends. In particular, the authors could be more clear on when they are referring to sparse connectivity vs sparse coding (be aware that “sparsening”, “sparseness” etc in this context can be ambiguous).

We have changed the terminology to be more consistent. We have clarified the text so that “sparsening” and “sparseness” are only used in conjunction with “activity” or “patterns”. We also now clarify that these measures relate to “population sparseness” rather than synaptic connectivity.

Also please specify when referring to input layer correlations vs output/ gc layer correlations.

We have modified the text to specify when we mean MF (input) correlations or GC (output) correlations.

Reviewer 2

This paper is a very thorough computational study of the expansion recoding idea in the context of feedforward circuitry, originally proposed by Marr and elaborated by Albus. There have been a number of papers over the past decades simulating cerebellar networks, including by this submission’s authors. Now this submission is likely the most exhaustive study yet of the specific problem of pattern separation in a feedforward network. While the results are based on the mossy fiber (MF) to granule cell (GC) synapse in cerebellum, and its modulation by Golgi cell inhibitory input, they are applicable to feedforward nets generally, where one suspects an expansion of the coding space in the presence of divergent connectivity.

Previous work in Silver’s group has shown lossless sparse encoding by sparse synaptic connectivity. This submission provides further insight into how the putative pattern separation works in terms of the tradeoff between sparseness and decorrelation caused by expansion of dimensionality and coding space (assessed with a measure of the variance of the activity patterns), and the nonlinear thresholding by the neurons. It does so by focusing on the question of why learning speed varies with the level of correlation between MF input patterns.

Separation is assessed by the speed of learning of a simple perceptron decoder at the output of the granule cells - which of course is a first step in cerebellar processing (this would then feed into the Purkinje fiber network, but that is not studied here). The results are obtained for a simple threshold linear model, as well as for a more biophysically detailed model.

The authors show that the increase in population sparseness going across the MF-GC synapse can not explain the learning speed’s dependence on input cross-correlation; rather, sparseness only aids pattern separation for independent inputs. For the more realistic correlated input pattern case, expansion of coding space using a large number of GCs and decorrelation caused by thresholding (i.e. nonlinearity that filters out subthreshold correlations) explain the behaviour of learning speed. This is the main result.

The model also shows that when network activity further affects thresholds - to approximate real feedforward inhibition from MF to Golgi cells - the decorrelation improves further still.

Most techniques used here are standard; of note is the novel way they randomly shuffle activity patterns all the while preserving population correlations, total variance and mean activity. The strength of the paper is really in bringing a lot of ideas together in a confined computational context, and disentangling certain effects behind expansion coding.

This computational study is thoroughly done and beautifully presented. It is very solid work. The supplementary material even looks at whether the story changes if other expansion ratios from mossy to granule change, since the glomerulus synapse is a very complex system and the ratio is not clear - this also allows them to generalize their findings to other feedforward systems with different ratios. The SI also looks at adaptive thresholding.

In spite of this praise for the work, in the end I am left with the uneasy recognition that, in spite of all this analysis, there are no real surprises and fundamental new revelations, and the sole focus on explaining how learning speed depends on input correlation is slightly narrow. Partial effects on which the story is built here have previously been described in the literature, such as the need to observe a large fraction of cells to say anything statistically significant about decorrelation.

We are glad that the reviewer appreciated the thorough analysis and clear presentation of our study. In response to the reviewer's comments on the significance of our findings, we have rewritten several parts of the manuscript to highlight the novelty of our work and to clarify the distinctions between our study and previous research. Towards this end we have also included three new supplementary figures.

We believe that our study provides a number of novel and important insights into the relationship between structure and function of feedforward networks, which are widespread across brain areas and animal species. While sparse coding, expansion recoding, and pattern decorrelation have all been identified as factors that contribute to pattern separation and learning in cerebellar-like systems, their relationship to network structure and their relative importance under different activity regimes is poorly understood. By combining statistical approaches and pattern separation analyses with a novel shuffling technique, we were able to isolate the effects of decorrelation from expansion and sparseness for the first time. Our finding that the key mechanisms underlying pattern separation are expansion and correlations, rather than sparse coding, overturns a long-standing assumption in cerebellar research.

Second, previous theoretical research has analyzed idealized, uncorrelated input patterns. However, it is known that MFs can fire over a wide range of different properties; and due to their fractured map organization, it is likely that

they are co-activated in spatially localized clusters. We extended previous work to study MF patterns with varying f_{MF} and spatial correlations. Our finding that the sparse synaptic connectivity observed in cerebellum and in cerebellar-like systems is optimal for pattern separation over the full range of MF statistics implies that it can act as a generic preprocessing unit, and provides a rationale for the homogenous structure observed throughout the cerebellar granular layer despite the diversity of MF properties.

Finally, previous work has studied pattern separation in discrete analytical models. We verify for the first time that pattern separation, decorrelation, and downstream learning can occur in a biologically constrained, detailed spiking model with noisy, Poisson MF firing.

For these three reasons, we believe our study represents a substantive advance from the previous work and thus constitutes a ‘fundamental new revelation’, to use the reviewer’s term. We thank the reviewer for pointing out that the novelty of our findings was unclear. We have modified the motivation in the Abstract, Introduction (p.4), and the Discussion (p.19-20) to emphasize these points.

We agree with the reviewer that just reporting that many cells are required to attain the statistical significance required to detect decorrelation would not be novel. However, we are making a different point here: our results show that the pairwise Pearson correlation coefficient *cannot be used in principle* to accurately measure decorrelation in these feedforward networks, regardless of the number of cells measured (and thus the statistical sampling). We introduce a novel measure, population correlation, which correctly assays correlations at the network level. The key difference is that the Pearson correlation coefficient only describes the activity of pairs of cells, while the population correlation has additional information about how those pairs together shape the distribution of the full population. This new approach is important, because pairwise Pearson correlation measures are currently widely used, potentially leading to an underappreciation of the extent of decorrelation in networks.

The ability of our measure of population correlation and the inability of the pairwise Pearson correlation coefficient to detect decorrelation is shown in new Supplementary Fig. 3, as well as revised Figure 3d, which averages over all pairs of cells in the full population (hence, all cells are observed) and over 25 trials, resulting in low standard error. The reviewer’s concern about significance may have been exacerbated by the original Figure 3d, which showed the results from a single trail, rather than being averaged over 25 trials like Fig. 3c. As a result, the correlation coefficient appeared to be much noisier than the population correlation.

We have updated Figure 3d in the revised manuscript accordingly. In addition, we have clarified this issue in the text (p. 12) and we have added a

supplementary figure (Supplementary Fig. 3) to show that our result is independent of statistical significance.

The recent Babadi-Sompolinsky paper, although cited, receives little further attention in the Discussion despite its focus on similar issues with clustered inputs.

While Babadi & Sompolinsky 2014 also describes 'clustered' inputs, their inputs are clustered in a different sense. They use input activity patterns that are clustered around a set number of random activity patterns in state space. On the other hand, we have analyzed MF activity patterns that are random in state space, but clustered in physical space, leading to positive structured correlations between MFs that are not present in the method employed by Babadi & Sompolinsky (see new Supplementary Fig. 6). The Babadi & Sompolinsky method is more similar to the case of independent MF patterns than to the MF patterns with spatially clustered correlations, as the average correlation coefficient is ~ 0 in their case. The presence of the spatial correlations present in biological networks substantially increases the difficulty of the classification task. To our knowledge, our study is the first to address the question of pattern separation of spatially correlated MF activation patterns. We have added a sentence clarifying the distinction between the two papers in the Methods (p. 26), as well as Supplementary Fig. 6.

The current submission interestingly provides a rationale for the observed 4 synapses per GC cell observed, as it led to the highest learning speed. Yet it is somewhat obvious that there has to be a moderate connectivity ratio, since connecting 1 to 1 makes the granule cells linear and incapable of decorrelating with thresholding, and a large ratio leads to too much correlation that is difficult to remove, especially for correlated inputs.

We agree that it is intuitive that there would need to be a moderate optimal connectivity for pattern separation. However, the finding that the theoretical optimum coincides with the observed value of ~ 4 synapses per GC, a number which has been evolutionarily conserved since the appearance of fish, is not obvious, given that this optimum is preserved for different expansion ratios (Supplementary Fig. 1) and in the presence of network activity dependent thresholding (Supplementary Fig. 2). Moreover, it is surprising that the optimal connectivity level is robust to changes in MF correlations. This result is significant because it shows that the granule cell layer could potentially perform pattern separation flexibly over a very broad range of input properties. We have modified the introduction and discussion to emphasize the importance of robust performance in response to diverse inputs (p.4, 21).

The main finding that sparseness is a weaker determinant of the learning speed vs input correlation does provide "insight", as the authors write, but in my mind does not constitute a sufficiently significant advance or breakthrough that is more required for NCOMM. While it is close to the bar, I consider the

work more suitable for other general neuroscience journals Scientific Reports, J. Neuroscience, eLife and the like.

In this study we quantified the contribution of different factors (expansion, sparse coding, correlations, and network structure) to pattern separation in divergent feedforward networks for the first time. Our results show that the synaptic connectivity within the cerebellar input layer is optimal for separating correlated input patterns, and that population sparseness is not a main determinant.

These new findings are timely as they provide a framework for understanding recently reported dense population coding in granule cells. Sparse coding has been an influential concept in the cerebellar literature for the last 50 years. However, two experimental studies published this year (Giovannucci et al. 2017, Knogler et al. 2017) have found that granule cell population activity may be considerably denser than previously thought, challenging this central tenet of Marr-Albus theory. By demonstrating that expansion and decorrelation, rather than sparse coding, are the main determinants of pattern separation, our results show that cerebellar input layer networks can perform efficient pattern separation irrespective of whether the GC population is sparse or dense. Indeed, we find that the sparsely connected feedforward networks found in the cerebellum and elsewhere can increase downstream learning speed even when the vast majority of GCs are active (new Supplementary Fig. 5). Thus, our results show that pattern separation can be effective under the high activity regimes reported by Giovannucci et al. who found that 2/3 of the observed granule cells were activated during the conditional stimulus, and Knogler et al., who found that over 50% of granule cells in immature Zebrafish were active in response to moving gratings and natural scenes.

For these reasons, we believe that our study represents a substantive advance in our understanding of information processing in cerebellar and cerebellar-like feedforward networks and will be of considerable interest to a wide range of scientists. We have rewritten the Discussion to clarify the importance of this finding and to relate them to these recently published papers (p.20). We also have added Supplementary Fig. 5 to quantify the GC population activity in regimes of effective pattern separation, thereby enabling our results to be set in the context of dense and sparse encoding.

Specific minor points:

p.7 When you mention that you first tested whether sparse coding could explain the dependence of learning speed on connectivity, remind the reader that this addresses Fig.1f.

We have added this reference to the text.

top of p.8: the sparseness effect actually goes in the opposite direction. So if I understand correctly, one should mention that, in fact, whatever does end up

explaining the increase in normalized learning speed with input cross-correlation has to also work against this loss of sparseness.

We have added this statement to p.8-9.

p.8 bottom: "...and the size of the coding space": I assume you meant "variance" instead of "size" here? You should make it clear whether you consider these terms interchangeable or not.

We used "normalized total variance", "expansion", and "size of the coding space" interchangeably in the original text. For simplicity we have now modified the text to refer only to the first two terms. We also now state the equivalence of "normalized total variance" and "expansion of coding space" explicitly on p.9.

p.10: when you mention Pearson coefficient, explain it more: emphasize that it is a pairwise measure, averaged over all (?) pairs, and still deals with a normalized GC/MF measure?

We have added these clarifications to the text and caption of Fig. 3.

p.13: Fig.5d: why not call this the number of inputs as in previous plots? (same comment for the ordinate of Fig.3c,d).

We have corrected the axes of Fig. 5d and Fig. 3c,d.

p.14 "In contrast, learning was enhanced (up to a 4-fold increase)...": where is this shown?

This was shown in Figure 5e. We have added this reference in the text.

p.16: "...this is likely due to the change in the GC input-output function..." : this is the second time one reads this reason in less than a page, and it refers to previous work by the authors. The reader needs to know what this change is to properly understand what is meant.

Billings et al. showed that as the number of synaptic connections increases, the nonlinearity in the GC input-output function becomes sharper. Therefore the effects that we predict from the analytical model that are due to the GC nonlinearity are in fact even stronger in the detailed model. We have clarified this in the text (p.17).

p.22: "Nevertheless, decorrelation of GC activity is expected to improve the performance of the conversion of spatial activity into temporal sequences via these mechanisms." This expectation needs to be clarified, it seems a big leap.

We meant that that decorrelation of GC activity is expected to also improve the performance of conversion of spatial activity into distinct temporal sequences, facilitating temporal pattern separation as suggested in Laurent 2002. However, due to length constraints and the additions we made to the Discussion on the advice of both reviewers, we decided to remove this section on temporal pattern separation.

p.25 top: typo: diameter ball of within the granular layer

We have corrected this typo.

p.27: specify: correlation matrix of what ?

The covariance matrix of either the MF or GC activity patterns. We have specified this in the text.

p.28 top: "...we swapped the spike counts for random pairs of patterns." : not clear what this means.

We mean that we took two random GC patterns, and exchanged the value of the neuron's spike count in one pattern with its spike count in the other. That step is then iterated many times until the desired correlation is achieved. This procedure has been clarified in the Methods (p.29).

In the figures and text, MF activity is never clearly defined. I assume it is, as usual, the fraction of active MF cells.

The MF activation level is defined as the fraction of active MF cells in a pattern. As stated in our response to Reviewer 1, we have modified the text to refer only to "fraction of active MFs," or f_{MF} (now defined on p.6), for consistency and clarity.

Fig.1b: tick marks are missing

We have corrected the figure.

Fig.3: specify whether you equate active decorrelation with negative normalized correlation.

Decorrelation is defined by the normalized population correlation being less than 1. Note that Fig. 3c,d shows the \log of the normalized correlation, so that decorrelation corresponds to negative values. We have clarified this in the figure caption and in the text (p.11).

REVIEWERS' COMMENTS:

Reviewer #1 (Remarks to the Author):

The authors have fully addressed my concerns. I congratulate them on an excellent study that provides fundamental insights about divergent feedforward networks. I strongly support publication in Nature Communications.

Reviewer #2 (Remarks to the Author):

The authors have done a wonderful job clarifying the novelty and potential impact of their work, and answering my other queries. The addition of the 3 supplemental figures is completely warranted as well and accentuates the novelty, especially the comparison between pearson coefficient and their population correlation measure and the difference with the Babadi method. I am happy to recommend publication.

I leave it to the authors to decide whether to change the words "expansion and correlations" to "expansion and decorrelation" in their abstract and Discussion, as the latter formulation more directly explains what is relevant to pattern separation.

Replies to the reviewers' comments

Reviewer 1

The authors have fully addressed my concerns. I congratulate them on an excellent study that provides fundamental insights about divergent feedforward networks. I strongly support publication in Nature Communications.

We thank the reviewer for his/her feedback and support.

Reviewer 2

The authors have done a wonderful job clarifying the novelty and potential impact of their work, and answering my other queries. The addition of the 3 supplemental figures is completely warranted as well and accentuates the novelty, especially the comparison between Pearson coefficient and their population correlation measure and the difference with the Babadi method. I am happy to recommend publication.

We thank the reviewer for his/her feedback and support.

I leave it to the authors to decide whether to change the words "expansion and correlations" to "expansion and decorrelation" in their abstract and Discussion, as the latter formulation more directly explains what is relevant to pattern separation.

We feel that it is more accurate to say "expansion and correlations" rather than "expansion and decorrelation". As we discuss in the Results, the avoidance of network-induced correlations, even without active decorrelation, is crucial for performance of sparsely connected networks. We have therefore left this wording in the abstract and discussion as is.